# Inflammation, Infiltration, and Evasion—Tumor Promotion in the Aging Breast

**DOI:** 10.3390/cancers15061836

**Published:** 2023-03-18

**Authors:** Nicole Cruz-Reyes, Derek C. Radisky

**Affiliations:** Department of Cancer Biology, Mayo Clinic, Jacksonville, FL 32224, USA

**Keywords:** breast cancer, aging, immune system, immunoediting

## Abstract

**Simple Summary:**

Breast cancer is a significant health problem affecting millions of women worldwide, particularly post-menopausal women. Although early detection and treatment have improved, many patients still develop metastatic disease. Recent research has focused on using the immune system to diagnose and treat breast cancer. The breast immune microenvironment regulates tissue homeostasis and resistance to tumor growth. This review examines how the immune system changes with age, how these changes affect breast cancer development and progression, and how targeted therapies that utilize the immune system can improve patient outcomes. It emphasizes the importance of understanding the relationship between aging, the immune system, and breast cancer, and the potential of immune-based therapies to combat this devastating disease.

**Abstract:**

Breast cancer is a significant cause of morbidity and mortality in women, with over two million new cases reported worldwide each year, the majority of which occur in post-menopausal women. Despite advances in early detection and treatment, approximately one-third of patients diagnosed with breast cancer will develop metastatic disease. The pathogenesis and progression of breast cancer are influenced by a variety of biological and social risk factors, including age, ethnicity, pregnancy status, diet, and genomic alterations. Recent advancements in breast cancer research have focused on harnessing the power of the patient’s adaptive and innate immune systems for diagnostic and therapeutic purposes. The breast immune microenvironment plays a critical role in regulating tissue homeostasis and resistance to tumorigenesis. In this review, we explore the dynamic changes in the breast immune microenvironment that occur with age, how these changes impact breast cancer development and progression, and how targeted therapeutic interventions that leverage the immune system can be used to improve patient outcomes. Our review emphasizes the importance of understanding the complex interplay between aging, the immune system, and breast cancer, and highlights the potential of immune-based therapies in the fight against this devastating disease.

## 1. Introduction: Understanding the Role of Aging and Menopause in Breast Cancer Development

Breast cancer is a significant health concern worldwide and is the leading cancer diagnosis in females. In the United States, breast cancer is responsible for an estimated 42,000 deaths each year [1]. While several risk factors have been linked to breast cancer, including a family history of the disease, specific genetic mutations, dense breast tissue, and previous exposure to radiation [2], advancing age is the most significant risk factor [3]. The majority of breast cancers are diagnosed after the age of 55 [4], and although tumors diagnosed later in life are typically less aggressive, the increased incidence in older women results in the highest overall number of breast cancer-associated fatalities.

The biological changes that occur during aging vary among different populations, and are influenced by genetics, environment, and lifestyle [5]. Menopause is a critical timepoint for women, occurring when ovaries stop producing hormones and menstruation ceases [6]. In Western countries, menopause occurs at an average age of 51 years, and approximately 85% of women have undergone menopause by age 55 [7,8]. Menopause leads to sudden decreases in estrogen and/or increases in cortisol levels, which have both immediate and long-term consequences on breast tissue and future breast cancer incidence. Recent research has highlighted the critical role of aging in the immune system of the breast as a significant risk factor for breast cancer. As women age, the coordination between the innate and adaptive immune systems diminishes, leading to tumor-immune equilibrium and eventual promotion of tumor growth [9]. Recognition of this process has prompted the development of agents that modulate and activate the immune system to target early and advanced stage breast cancers, and many of these agents are currently undergoing clinical trials. This review will examine how changes in the aging breast contribute to the breakdown of immunoediting and the transition to a tumor-promoting immune microenvironment, with a specific focus on how these changes are accelerated by menopause, how they occur in coordination with aging-related changes in the breast, and how they may be targeted for therapeutic benefit.

## 2. Harnessing the Immune System to Fight Cancer: From Early Observations to Personalized Medicine

Multiple anecdotal reports dating back to ancient Egypt and the early nineteenth century describe spontaneous tumor regression after an infection with concomitant high fever [10]. Two German physicians, Fehleisen and Busch, noticed significant tumor regression after erysipelas infection, making them the first to attempt to modulate patients’ immune systems to cure cancer [11]. However, their attempts to replicate the phenomenon proved unsuccessful until Fehleisen identified the bacterial strain responsible for erysipelas and tumor shrinkage as *Streptococcus pyogenes*. The formal beginning of immunotherapy dates to 1891, when William Bradley Coley discovered 47 case reports of cancer patients undergoing spontaneous remission after concomitant acute bacterial infection [12]. Further experimental evidence confirmed that robust activation of the immune system by infection enabled it to recognize and target tumor cells, as observed when treating sarcoma patients with a mixture of live and inactivated *Streptococcus pyogenes* and *Serratia marcescens* resulted in tumor regression [12]. Although “Coley’s toxins” were commercially available from 1899 and proved successful, their lack of a known mechanism of action and the high risk of infecting cancer patients with pathogenic bacteria led oncologists to prefer surgery and radiotherapy [13,14]. Furthermore, when researchers attempted to translate these findings to human trials, they faced significant challenges. The early trials were largely unsuccessful, and as a result, the idea of cancer immunotherapy was largely abandoned for several decades until the discoveries in the mid-twentieth century of key elements of the immune system, including interferon [15]. As a more recent example of this approach, the use of Interferon-alfa 2 to stimulate the immune system to attack cancer cells was approved by the FDA in 1986 for the treatment of hairy cell leukemia, and has since been successfully used to treat other types of cancer, such as chronic myeloid leukemia, malignant melanoma, and renal cell carcinoma [16]. Ongoing clinical trials are evaluating the efficacy of interferon-alfa 2 in other types of cancer, such as ovarian cancer and lung cancer. Based on these early findings, several cancer immunotherapies have been developed, including checkpoint inhibitors, CAR-T cell therapy, and vaccines.

However, as the complex roles of the immune system in both inhibiting and activating tumor progression have been revealed, it has become increasingly clear that specific components of the immune system must be regulated when designing immunotherapies, considering the patient’s unique immune status. This is particularly relevant when considering aging, as immune function declines with age, resulting in decreased cancer immunosurveillance and increased risk of cancer development. The concept of “cancer immunoediting”, which explains the immune system’s ability to control and shape the development of cancer, is also related to this.

Moreover, the role of the tumor microenvironment in shaping the immune response to cancer is important. While cancer immunotherapies have shown promising results, potential side effects and challenges, such as autoimmunity and resistance to treatment, have been observed. Therefore, there is an increasing focus on personalized medicine and the development of immunotherapies tailored to the specific characteristics of a patient’s tumor, tumor microenvironment, and immune system status. Age is a crucial consideration that can impact all of these factors.

## 3. Advances in Cancer Immunotherapy: Targeted Therapies and Emerging Approaches

The PD-1/PD-L1 interaction is a critical pathway in the regulation of the immune response. PD-1 (Programmed cell death protein 1) is a receptor expressed on the surface of T cells, while PD-L1 (Programmed death-ligand 1) is a ligand expressed on the surface of various cells, including cancer cells. This interaction helps to prevent autoimmunity by preventing the immune system from attacking healthy cells [17,18]. However, in some cancers, increased expression of PD-L1 inhibits immunoediting through exhaustion of antitumor cytotoxic T cells, leading to the cancer cells evading immune detection and destruction [17,19,20] (Figure 1A). Targeting the PD-1/PD-L1 interaction with immune checkpoint inhibitors, such as pembrolizumab and atezolizumab, can help to reactivate the immune response against cancer cells and improve the efficacy of cancer treatments. Pembrolizumab is an anti-PD-1 antibody that blocks the ability of PD-1 to inhibit T-cell activation [21] (Figure 1B), while atezolizumab selectively binds to PD-L1 to stop the PD-1/PD-L1 interaction [22,23,24] (Figure 1C). Under normal circumstances, Pembrolizumab and atezolizumab target the repressed immune system to re-activate the antitumor immune response [25]. This approach differs from conventional cytotoxic treatments that target cancer cell proliferation and is also resilient to mutational changes in the tumor itself [26,27,28,29,30]. In addition, the effectiveness of immune checkpoint inhibitors, such as pembrolizumab and atezolizumab, may be influenced by the immunological characteristics of the tumor. Immunologically hot tumors with high levels of immune cell infiltration and high mutational load are more likely to respond to checkpoint inhibitors by reactivating the antitumor immune response. In contrast, immunologically cold tumors with low levels of immune cell infiltration and low mutational load are less likely to respond to checkpoint inhibitors alone. Combination therapies that can convert cold tumors into hot tumors by increasing tumor antigen presentation, stimulating immune cell infiltration, or altering the tumor microenvironment are being investigated to improve the efficacy of checkpoint inhibitors in these tumors. Overall, targeting the PD-1/PD-L1 interaction with immune checkpoint inhibitors offers a promising approach to cancer treatment, particularly in immunologically hot tumors.

In a phase 3 clinical trial, atezolizumab in combination with nanoparticle albumin-bound (nab)-paclitaxel prolonged progression-free survival in patients with advanced triple negative breast cancer (TNBC). This treatment showed better efficacy and decreased toxicity compared to standard paclitaxel [31]. In 2019, the FDA approved the use of atezolizumab and nab paclitaxel for PD-L1 positive unresectable locally advanced or metastatic TNBC and also approved the use of the VENTANA PD-L1 assay as a companion diagnostic device for selecting TNBC patients for atezolizumab treatment [32,33,34,35]. In 2020, the FDA approved a combination immunotherapy for advanced TNBC consisting of pembrolizumab and chemotherapy [36]. This combination therapy has been shown to result in significantly longer overall survival than chemotherapy alone [37]. In 2021, it was approved for high-risk, early-stage TNBC as well [38]. Other immune checkpoint inhibitors are also being tested for breast cancer. Durvalumab, another anti-PD-L1 antibody, is being studied in combination with chemotherapy in patients with HER2-negative metastatic breast cancer in NCT03356860. Nivolumab, another anti-PD-L1 antibody, is being studied in combination with ipilimumab in patients with advanced or metastatic solid tumors, including breast cancer in NCT01928394. Tremelimumab, an anti-CTLA-4 antibody, is being studied in combination with durvalumab in patients with advanced solid tumors, including breast cancer, in NCT02658214. The majority of clinical trials with immune checkpoint inhibitors have focused on TNBC, because this subtype is immunologically hotter than the other subtypes. However, there are a few clinical trials focusing on HER2-positive patients, including NCT04740918, which is evaluating Atezolizumab in combination with Trastuzumab Emtansine, and NCT04789096, which is evaluating Pembrolizumab in combination with two different HER2 inhibitors. The I-SPY2 clinical trial is evaluating the effect of Pembrolizumab plus neoadjuvant chemotherapy in women with early-stage ER-positive breast cancer, and the effect of Pembrolizumab with endocrine therapy is being evaluated in NCT02778685. These efforts continue to spur further research into the development of targeted immunotherapies for breast cancer with minimal side effects.

However, it should be noted that immunomodulatory treatments have specific side effects, including lung inflammation, colitis, and liver damage [39,40]. Additionally, they are currently quite expensive, costing approximately USD 150,000 per year, and are not covered by some national healthcare or insurance plans, rendering them unavailable for many populations. Future developments to suppress these pathways at lower cost and with reduced side-effects could have significant benefit.

CAR-T cell therapy is an emerging cancer immunotherapy approach that involves extracting the patient’s own T cells, genetically modifying them to attack the cancer, and then reintroducing the altered T cells back into the patient’s body (Figure 2). While CAR-T cell therapy has proven effective in hematologic malignancies and has been FDA approved to treat such cancers [41,42,43], it has yet to demonstrate the same level of success in treating solid tumors. This is partly due to several characteristics of solid tumors, such as their heterogeneity, toxicities, a hostile tumor microenvironment, and limited infiltration of immune cells [44,45,46]. Nevertheless, research is progressing towards identifying unique antigens in solid tumors that can be targeted by CAR-T cells [47]. In TNBC, some promising targets include TROP2 [48], GD2 [49], ROR1 [50], MUC1 [51,52], and EpCAM [53,54]. These antigens not only have high expression in breast cancer [47], but have also shown promising results in CAR-T experimental models. In addition, fourth generation CAR-T cells (TRUCKS, T-cell redirected for universal cytokine-mediated killing) are being developed to improve the effectiveness of CAR-T cell therapy for solid tumors [55]. While many clinical trials are currently exploring the use of different CAR-T cells for the treatment of breast cancer, results are still under review.

Cancer vaccines stimulate the immune system to recognize and attack cancer cell antigens by introducing specific antigens to activate both T and B cell responses (Figure 3). In advanced-stage ERBB2-positive breast cancer patients, several vaccines targeting ERBB2 have shown promise [56,57], although early attempts targeting this protein resulted in only “short-lived peptide-specific immunity” [58,59,60]. However, a phase 1 clinical trial of a vaccine targeting the intracellular domain resulted in the generation of persistent levels of ERBB2 type 1 T cells [56]. This vaccine has been associated with the induction of immunity at every dose tested, and it has now progressed to phase 2 clinical trials [61]. These promising results demonstrate the potential for effective cancer vaccines and how these could offer a novel treatment option for patients with advanced-stage ERBB2-positive breast cancer.

Recent studies have raised concerns about the efficacy and safety of cancer immunotherapy in older patients. Experiments evaluating cancer immunotherapy efficacy are generally conducted in young animals [62]. The effectiveness of cancer immunotherapy in elderly patients, who often experience immunosenescence (a decline in immune system function), may be limited. In addition, the elderly population is often burdened by age-related comorbidities, which may reduce the potential benefits of immunotherapy or increase the risk of adverse events. Therefore, careful assessment of the risks and benefits of cancer immunotherapy in elderly patients is warranted, considering their individual health status and potential limitations of their aging immune system. Strategies to enhance the efficacy and safety of immunotherapy in elderly patients, such as using lower doses, alternative dosing schedules, or combination therapy, should be further explored in clinical trials. It is important to ensure that cancer immunotherapy is accessible to all patients, regardless of their age, and to develop personalized treatment strategies that account for the unique needs and challenges of aging breast cancer patients.

## 4. The Impact of Age-Related Immunosenescence on Cancer Development and Potential Therapeutic Approaches

Age has complex effects on cancer development, including changes in the immune system that can promote tumor growth [63,64,65,66,67]. As we age, intrinsic anticancer activity becomes less effective and tumor-promoting functions develop. This is partially due to a decrease in the effectiveness of the immune system, leading to slower response times and increased risk of illness [68,69]. Age-associated dysregulation of hematopoiesis leads to a decrease in immune cell production [70] and age-related changes in the lymphoid organs, particularly affecting T cells and natural killer cells, which can also contribute to the reduced effectiveness of the immune system in older adults [71,72]. Age-related changes in the cytokine milieu can also impact the ability of the immune system to respond to cancer cells [73]. The aging process also affects the bone marrow microenvironment, leading to a decline in the function of hematopoietic stem cells (HSCs) and a decrease in the production of immune cells [74]. The accumulation of pro-tumorigenic myeloid cells in the bone marrow microenvironment can further exacerbate the age-related changes in the immune system [75]. Additionally, age-associated accumulation of fibrosis [76] and a decline in the function of dendritic cells can also affect the ability of the immune system to respond to cancer cells. These age-related dysfunctions of the immune system are referred to as ‘immunosenescence’ and result in delayed wound healing and impaired detection and elimination of unhealthy cells [77].

Age-associated immunosenescence has a more pronounced effect on the adaptive immune system. Changes in primary lymphoid organs, such as thymus and bone marrow, are responsible for immunosenescence-associated changes in adaptive immune cells [78,79]. The thymus is responsible for the production and maturation of T cells. With age, the thymus undergoes an age-related process called involution, where it becomes less active and is gradually replaced by fatty tissue [80]. This leads to a decrease in the production of new, naïve T cells with the capability of recognizing and responding to new antigens. The bone marrow, which is responsible for the production of B cells, also shows age-dependent declines in cellularity and replacement with fatty tissue [81,82]. This reduces the capacity of the immune system to produce new B cells and respond to new cancer-specific antigens.

Age-associated decline in primary lymphoid organ function can be delayed through healthy lifestyle choices, such as maintaining a healthy diet, regular physical activity, and avoiding smoking. Pharmacological approaches have also been studied to delay the decline in primary lymphoid organ function, including growth hormones and dehydroepiandrosterone (DHEA) [83]. However, once T cell producing thymic tissue has been completely lost, the exogenous administration of hormones will not enhance T cell production. Recent studies have suggested that a more promising strategy could be the sustained rejuvenation of primary lymphoid organs by targeting both thymic epithelial cells (TECs) and lymphoid cells [84]. While regenerative medicine techniques, such as stem cell therapy or tissue engineering, may hold potential for regenerating thymic tissue and enhancing T cell production [85], these methods may have limitations, and there may be safety and long-term efficacy concerns, as well as high costs associated with them. Nevertheless, the use of such approaches may help to attenuate the involution of the thymus and augment remaining thymus function to increase T cell levels [69], thus, potentially reducing the complications of immunosenescence.

Tumor progression is accompanied by the progressive accumulation of mutations. These mutations can manifest as neoantigens that can be identified and eliminated through immunoediting. However, the immunoediting function can become less effective with immunosenescence. Moreover, tumors can also develop mechanisms to suppress immunoediting, such as the expression of PD-1 (Figure 1A), allowing more cancer cells to survive and grow. Age-associated dysfunction of the immune system and suppression of immunoediting contribute to an increased risk of cancer development in aging adults.

## 5. The Link between Inflammation and Genomic Instability

Aging is a complex process that involves an increase in sustained inflammatory responses, which can contribute to the development of cancer [86]. Chronic low-grade inflammation and acute intermittent inflammation are two forms of chronic inflammation and involves the release of chemicals such as histamine, bradykinin, and prostaglandins by mast cells [87]. These signaling molecules trigger a cascade of events that are intended to combat and destroy foreign pathogens; however, they can also damage healthy cells. Pro-inflammatory cytokines such as TNF-alpha and IL-1beta have been shown to play an integral role in promoting cancer development [88]. Furthermore, chronic inflammation results in the accumulation of reactive oxygen species (ROS), which can damage DNA, cause mutations, and, ultimately, promote cancer [89].

The link between DNA damage-induced genomic instability and inflammation creates a self-sustaining loop that can contribute to the development and progression of cancer. The loop induces the expression of type I IFNs and other cytokines through the cGAS-STING pathway, thereby inducing inflammatory responses [90]. In addition, inflammation also promotes the growth and progression of cancer by promoting angiogenesis, stimulating cellular proliferation, and evading immune surveillance [91]. Chronic inflammation can lead to tissue damage and fibrosis [92], creating a microenvironment that is conducive for the growth and spread of cancer.

Chronic infections that lead to inflammation can significantly increase the risk of cancer development [93]. Human papillomavirus (HPV), a sexually transmitted virus that infects the epithelial cells of the cervix, vagina, penis, and anus, is one such chronic infection [94]. High risk strains of HPV, such as HPV 16 and 18, can cause persistent infections and lead to changes in the DNA of infected cells, which lead to cancer over time [95]. It is important to note that HPV can affect both men and women, and as such, HPV vaccination is recommended for both boys and girls to help prevent the spread of the virus and reduce the risk of HPV-related cancers. Helicobacter pylori (H. pylori), a bacterium that infects the stomach, can cause chronic inflammation in the stomach lining and produce cytotoxins that damage the DNA of stomach cells, increasing the risk of cancer development [96,97]. Chronic Hepatitis B and C viral infections can damage the liver, leading to chronic inflammation and risk of liver cancer [98,99]. Antiviral and vaccine-based strategies, as well as interventions to attenuate inflammation, have shown significant benefits as cancer prevention strategies [100,101,102,103,104,105,106]. In addition, exposure to environmental toxins. such as polycyclic aromatic hydrocarbons, polychlorinated biphenyls, and heavy metals, can increase the risk of cancer development in individuals with underlying chronic viral infections and chronic inflammation [107], making interventions to reduce exposure to these toxins essential for reducing the burden of cancer.

To mitigate the risks of inflammation-related cancer, it is important to adopt healthy lifestyle practices, such as maintaining a healthy diet, regular physical activity, and refraining from smoking. Anti-inflammatory drugs such as non-steroidal anti-inflammatory drugs (NSAIDs) and cyclooxygenase-2 (COX-2) inhibitors are pharmacological approaches that can attenuate chronic inflammation, but long-term use may have adverse side effects, especially in aging populations [108,109]. Alternatively, natural compounds with anti-inflammatory effects, such as curcumin, resveratrol, and epigallocatechin gallate (EGCG), have shown promising results [110]; however, more research is needed to determine the efficacy and safety of these compounds. By understanding the link between inflammation and cancer, and taking a proactive approach to reduce chronic inflammation, we can work towards a healthier, cancer-free aging population.

## 6. Targeting the Breast Immune Microenvironment for Breast Cancer Prevention

Modifying the immune system represents a potential strategy for preventing cancer development, and this approach can complement the development of immunotherapies [111]. Although primary prevention for breast cancer has some limitations in terms of safety and minimal toxicity in women prior to disease development, the use of vaccines in this setting can be a promising approach. However, this requires the identification of antigens associated with and/or upregulated in high-risk lesions leading to breast cancer [112]. A promising population for prevention strategies is women with benign breast disease (BBD), a condition that is known to increase the risk of breast cancer. Studies of women with BBD have revealed changes in the immune system that occur before cancer development, indicating that tissue abnormalities may promote inflammation while decreased immune responses may precede tumor development. Specifically, Degnim et al. reported that women with BBD have increased immune cell presence, including T cells, macrophages, B cells, and dendritic cells, but decreased immune cell numbers in those who later develop breast cancer [113]. Furthermore, immune infiltration patterns in breast cancer can influence clinical outcomes. Raza Ali et al. reported that CD8+ T cells were associated with a reduction in the risk of relapse in ER- tumors, although an increase in T helper cells were indicative of better response to neoadjuvant chemotherapy; by contrast, infiltrating macrophages were associated with poor prognosis in ER+ tumors [114]. Finally, increased T regulatory cells were associated with poor prognosis, regardless of ER expression. Identifying the immune cell populations upregulated in BBDs that progress to cancer and using vaccines to target and activate the immune system may offer a helpful preventive strategy for breast cancer.

Advancements in anticancer vaccine therapies have demonstrated potential for their use as primary prevention or prevention after remission. However, identifying the most effective vaccine targets is essential for success. A phase I clinical trial is currently studying the safety and toxicity of a vaccine to prevent recurrence of cancer in patients with non-metastatic, node positive, HER2 negative breast cancer who are in remission [115], and vaccine strategies are being tested in patients with HER2+ ductal carcinoma in situ (DCIS) [116]. These advancements demonstrate the potential for vaccines to be used as a primary prevention strategy for breast cancer, but more research is needed to identify the most effective vaccine targets. For vaccines to be a feasible prevention therapy for healthy patients, they must have minimal side-effects and deliver sustained immunological response. Achieving this goal is likely to require antigens that trigger combined CD4/CD8 activation. Additionally, vaccines that target multiple cancer antigens simultaneously are expected to have better efficacy in preventing cancer [112], although effective target selection will need to consider patient age, as gene expression varies significantly by age in breast cancer patients [7]. The development of cancer vaccines is a complex and challenging process that requires significant resources and expertise, particularly if planned for the prevention setting. The support of funding agencies, pharmaceutical companies, and the wider scientific community is crucial for the continued progress of this field.

Breast cancer is a complex disease, and understanding the changes that occur in the breast microenvironment is essential for developing accurate interventions for prevention. Recent research has shed light on the human breast microbiome and its association with prognosis. Tzeng et al. found that certain bacteria such as Anaerococcus, Caulobacter, and Streptococcus were present in benign breast tissue, while Propionibacterium and Staphylococcus were depleted in tumors and correlated positively with T cell activation-related genes [117]. These findings suggest that the microbiome in healthy breast tissue is different from breast cancer, and knowing these differences is helpful in understanding onset and progression of the disease [118]. Meanwhile, Hieken et al. evaluated stromal fat and fibrosis percentage in benign breast tissue and breast cancer and found that fat percentage was lower and fibrosis percentage was higher in samples from patients with benign disease versus cancer [119]. Ultimately, understanding the dynamic changes in the breast immune microenvironment can aid in the modification of tumor-promoting factors and the development of effective prevention strategies.

## 7. The Complex Interplay between Menopause, Microbiome, and Breast Cancer

Breast cancer incidence is strongly associated with aging, and menopause is a significant event in women’s lives that brings with it a range of physical symptoms [120]. These symptoms include hot flashes, sleep disruption, anxiety, depression, and decreased energy levels [121]. Post-menopausal women experience changes in the immune system attributed to estrogen deprivation [122]. The rapid reduction in estrogen levels causes declines in immunologic reactivity due to decreased numbers of CD4 T and B cells, decreased cytotoxic activity of NK cells, and increased production of proinflammatory cytokines [123], which may increase the risk of breast cancer.

Recent research has shed light on the complex interplay between the aging breast, immune system, gut microbiome, and breast microenvironment during the menopausal transition. The gut microbiome is increasingly recognized as a key modulator in breast cancer development [124], and it has been shown to change during menopause, with alterations in both the diversity and abundance of microbial taxa [125,126]. Such changes have been attributed to hormonal fluctuations, alterations in diet, and shifts in the immune system. Emerging evidence suggests that microbiome complexity extends beyond the gut, with variations in microbial communities detected in the breast and the nipple discharge of women [127].

While recent studies have produced somewhat conflicting results, several have suggested that microbial dysbiosis may be associated with an increased risk of breast cancer in post-menopausal women. For example, a study published in 2018 found that microbial composition differed between post-menopausal women with breast cancers compared to controls, whereas no such difference was observed in pre-menopausal women with breast cancer [128]. However, a 2021 study of post-menopausal women with ER+/HER2- breast cancer reported no significant differences in microbiota richness, composition, or diversity between these women and those with negative mammography results [129]. These discrepancies may reflect factors such as previous neoadjuvant treatments and antibiotic use, which can significantly impact microbiota and, thus, confound results. Another review published in 2022 reported conflicting results in post-menopausal women regarding the strains of bacteria that were increased or decreased in these groups [130]. A more in-depth 2022 study showed that the microbial composition of normal adipose breast tissue in breast cancer patients differed, and specific breast bacterial profiles were associated with tumor characteristics [131]. These findings suggest that the microbiome may not only be a biomarker of cancer development but could also potentially contribute to the maintenance of the tumor microenvironment [132].

The growing body of research on age-associated changes in the microbiome and its impact on breast cancer underscores the importance of taking a more holistic approach to understand the complex biological processes that drive cancer development and progression. By elucidating the links between the gut microbiome, aging, and breast cancer, researchers can begin to identify new targets for prevention and treatment strategies, ultimately leading to improved outcomes for patients. Further studies are needed to explore the interplay between the microbiome and other factors such as genetics, lifestyle, and hormonal changes, with the aim of providing more personalized approaches to breast cancer prevention and treatment.

## 8. Conclusions

Aging is a complex and inevitable process that is associated with numerous changes in the body, including an increased risk of breast cancer. Menopause and the resulting hormonal changes, immune senescence, and changes in the microbiome all play a significant role in the development and progression of breast cancer. However, recent advances in our understanding of the immune system have paved the way for the development of novel anticancer therapeutic approaches. Immune checkpoint inhibitors have been developed to reactivate the immune system to target cancer cells more effectively. Adoptive T cell transfer has also been employed to enhance the immune system’s ability to recognize and destroy cancer cells, while cancer vaccines have been developed to selectively activate the immune system to recognize and target cancer cells, even prior to cancer development. Additionally, the potential role of the microbiome in cancer development and progression is an emerging field of research, leading to investigations into microbiome-based interventions for cancer prevention and treatment. Additionally, it is important to note that the tumor cells and their microenvironment, which include the immune system, co-evolve as the tumor progresses and responds to therapy [133].

Despite these promising developments, much work remains to be carried out. Further research is needed to fully elucidate the complex interplay between aging, the immune system, and cancer development. Additionally, there is a need to develop more personalized approaches to cancer prevention and treatment that consider individual differences in genetics, lifestyle, and the microbiome. Nevertheless, the increasing understanding of the immune system and the microbiome provides new avenues for the prevention and treatment of breast cancer, with the potential to significantly improve patient outcomes.

## Figures and Tables

**Figure 1 cancers-15-01836-f001:**
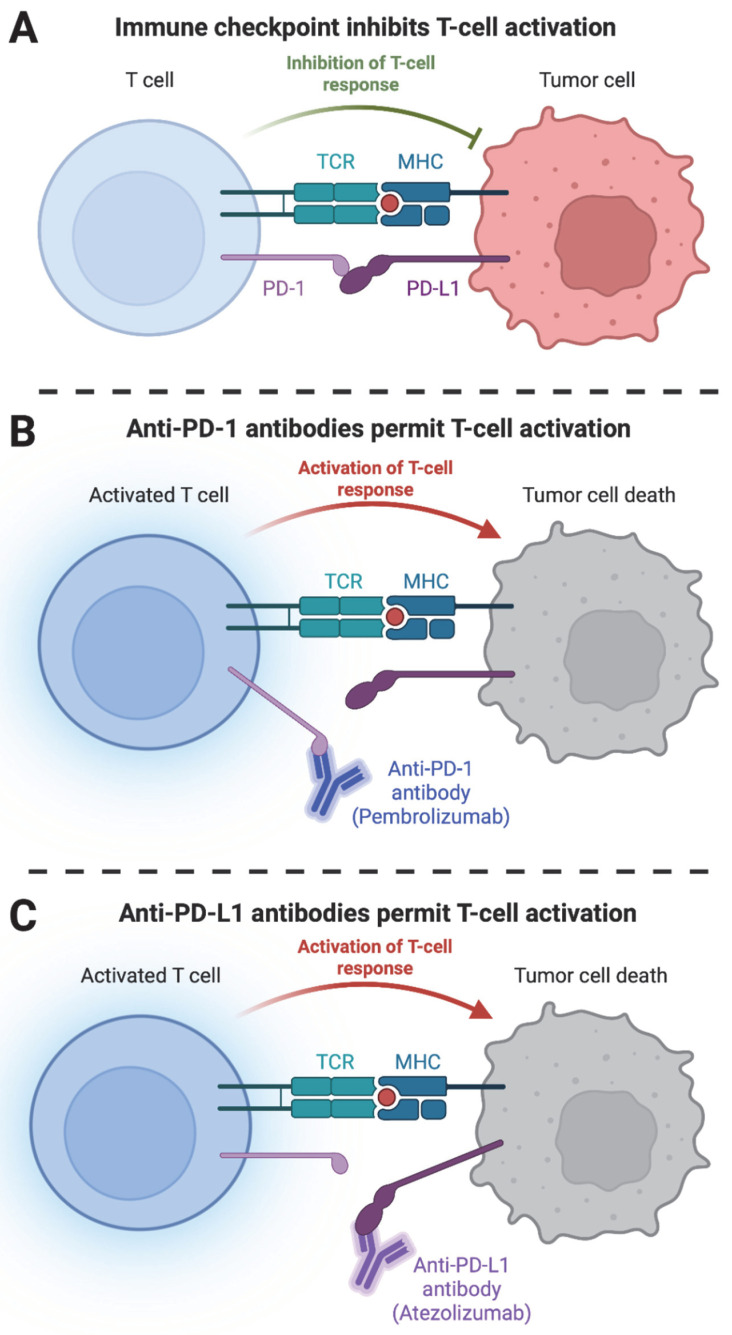
PD-1/PD-L1 function and targets. (**A**) Immune checkpoint inhibits T-cell activation and prevents tumor cell death. (**B**) Anti-PD-1 and (**C**) anti-PD-L1 antibodies (also known as immune checkpoint inhibitors) activate T-cell response and cause tumor death.

**Figure 2 cancers-15-01836-f002:**
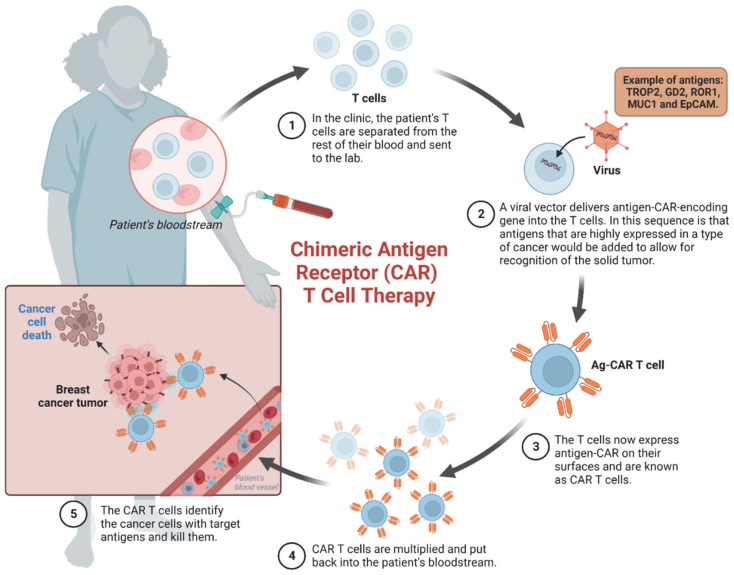
Chimeric Antigen Receptor (CAR)-T Cell Therapy for breast cancer. Step-by-step process of CAR-T cell therapy development.

**Figure 3 cancers-15-01836-f003:**
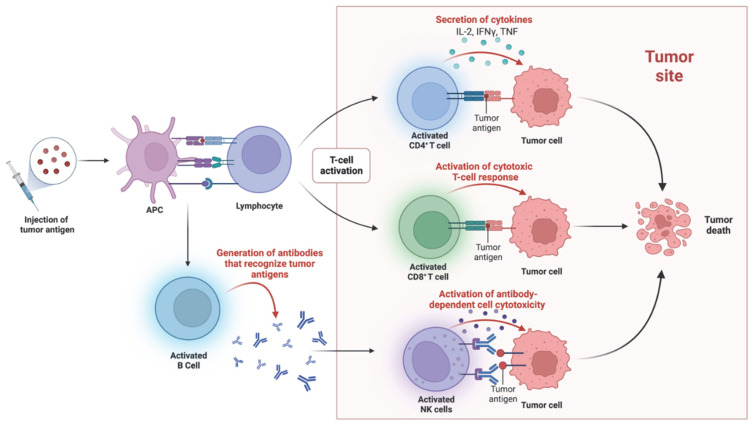
Cancer Vaccines Activate the Immune System to Target Tumor Cells. Cancer vaccines introduce tumor antigens (proteins found on the surface of cancer cells) into antigen-presenting cells (APCs), such as dendritic cells, which then activate CD8+, CD4+, and B cells to mount an immune response against cancer cells. CD8+ T cells, also known as cytotoxic T cells, are responsible for killing cancer cells directly. CD4+ T cells, also known as helper T cells, help activate and coordinate the immune response by releasing cytokines that attract other immune cells to the site of the tumor. B cells, another type of immune cell, produce antibodies that can target cancer cells. Antibodies can bind to tumor antigens on the surface of cancer cells, marking them for destruction by other immune cells or activating complement proteins that lead to cell death.

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
