# Peer review of "Inflammation, Infiltration, and Evasion—Tumor Promotion in the Aging Breast"

_cancers, 2023, doi:10.3390/cancers15061836_

Round 1
Reviewer 1 Report
In this review, the authors aimed to emphasize importance of understanding the complex interplay between aging, the immune system, and breast cancer. Several points should be noted as below.
1) Many papers has studied the “aging” in breast cancer. So, how “aging” in defined, it is equal to menopause mentioned in this paper?
2) The issue is “the Aging Breast”, as indicated in the title. However, except for the part “4. The Impact of Age-Related Immunosenescence on Cancer Development and Potential Therapeutic Approaches”, other parts seem to have nothing to do with Aging breast.
3) Immune cell states change, and the “hot” and “cold” state of the tumor also change. Talk something about it.
4) The “aging and immune response in breast cancer prognosis” should be provided more details.
5) How about the impact of aging-related immune on breast cancer stroma?
6) Part 5 was missing.
7) Figure 1 as to “PD-1/PD-L1 function and targets.” is well known. So, how about aging-related “PD-1/PD-L1 function and targets.”?
8) An interesting paper has recently proposed that “cancer as multidimensional spatiotemporal “unity of ecology and evolution” pathological ecosystem” that cancer cells dynamically interplay with their specific micro-/environments such as microorganisms, immune cells, stromal cells, adapt to each other and even co-evolve in a cross-level manner.(Luo W. Nasopharyngeal carcinoma ecology theory: cancer as multidimensional spatiotemporal “unity of ecology and evolution” pathological ecosystem. Theranostics 2023; 13(5):1607-1631. doi:10.7150/thno.82690. https://www.thno.org/v13p1607.htm). Such view should be updated.
Author Response
1) Many papers has studied the “aging” in breast cancer. So, how “aging” in defined, it is equal to menopause mentioned in this paper?
We thank the reviewer for this opportunity for clarification. The final sentence of the introduction, on page 2, lines 57-61 makes this point clear: “This review will examine how changes in the aging breast contribute to the breakdown of immunoediting and the transition to a tumor-promoting immune microenvironment, with a specific focus on how these changes are accelerated by menopause, how they occur in coordination with aging-related changes in the breast, and how they may be targeted for therapeutic benefit.”.
2) The issue is “the Aging Breast”, as indicated in the title. However, except for the part “4. The Impact of Age-Related Immunosenescence on Cancer Development and Potential Therapeutic Approaches”, other parts seem to have nothing to do with Aging breast.
We note that aging is mentioned or referred to in: chapter 3 ~lines 237, where the concern about efficacy and safety of cancer immunotherapy in older patients is raised, and discussed through the last paragraph of that section. Chapter 5 also opens discussing inflammation and genomic instability as a result of aging. Chapter 6 also compares the differences of immune microenvironment in young versus aged people. Chapter 7 introduces menopause as a “step” in the aging continuum, and suggests more research is needed to look at age-associated changes for the breast.
3) Immune cell states change, and the “hot” and “cold” state of the tumor also change. Talk something about it.
In chapter 3, a section was added to address this tumor stage: “In addition, the effectiveness of immune checkpoint inhibitors, such as pembrolizumab and atezolizumab, may be influenced by the immunological characteristics of the tumor. Immunologically hot tumors with high levels of immune cell infiltration and high mutational load are more likely to respond to checkpoint inhibitors by reactivating the antitumor immune response. In contrast, immunologically cold tumors with low levels of immune cell infiltration and low mutational load are less likely to respond to checkpoint inhibitors alone. Combination therapies that can convert cold tumors into hot tumors by increasing tumor antigen presentation, stimulating immune cell infiltration, or altering the tumor microenvironment, are being investigated to improve the efficacy of checkpoint inhibitors in these tumors. Overall, targeting the PD-1/PD-L1 interaction with immune checkpoint inhibitors offers a promising approach to cancer treatment, particularly in immunologically hot tumors.”
We also added this reference: 10.3389/fimmu.2021.682435
4) The “aging and immune response in breast cancer prognosis” should be provided more details.
Chapter 6 provides insights into breast cancer prognosis, as a result of immune cell populations.
5) How about the impact of aging-related immune on breast cancer stroma?
In chapter 6, immune cells in breast cancer stroma is mentioned. Reference used: Hieken, T.J., et al (2022).
6) Part 5 was missing.
Thank you for your attention and noticing this, we fixed it.
7) Figure 1 as to “PD-1/PD-L1 function and targets.” is well known. So, how about aging-related “PD-1/PD-L1 function and targets.”?
Besides senescence, there is little information about this and we agree that it is a very interesting topic for future research.
8) An interesting paper has recently proposed that “cancer as multidimensional spatiotemporal “unity of ecology and evolution” pathological ecosystem” that cancer cells dynamically interplay with their specific micro-/environments such as microorganisms, immune cells, stromal cells, adapt to each other and even co-evolve in a cross-level manner.(Luo W. Nasopharyngeal carcinoma ecology theory: cancer as multidimensional spatiotemporal “unity of ecology and evolution” pathological ecosystem. Theranostics 2023; 13(5):1607-1631. Doi:10.7150/thno.82690. https://www.thno.org/v13p1607.htm). Such view should be updated.
We thank the reviewer for this reference. It has been added along with new text on p. 12, lines 477-479: “Additionally, it is important to note that the tumor cells and their microenvironment, which include the immune system, co-evolve as the tumor progresses and responds to therapy [101].”
Reviewer 2 Report
Dear Authors,
first of all - congratulations for your great effort and interesting topic of the paper. I hope that my hints will be helpful and your final paper will be of great value to the scientific community. I really enjoyed reading it - let's make it clear.
- Despite the well-rooted habit of using abbreviations in Medicine sometimes too many such shortcuts make it more difficult to understand instead of facilitating the topic. It might be worth considering to only use well-known shortcuts, but not to create too many new ones.
- In some paragraphs formatting should be improved, as well as double spaces etc.
- Chapter 2: the history of immunotherapy starts a bit earlier, it may be good to highlight it; there are several good reviews in this regard, e.g. 0.3389/fimmu.2019.02965
- Lines 72-72: the beginnings, ok, but why those early experiments were stopped? Why resumed decades later?
- Line 86: I believe "increased RISK OF cancer development"?
- Figure 1: there are many more antibodies anti PD-1/PD-L1 currently, both FDA-approved and under clinical trials; it should be good (from a scientific point of view) to mention them, not solely pembro/atezo; this may suggest you're being paid by those pharma companies and we don't want to lose your scientific objectiveness.
- Figure 2 and 3 - size should be changed to fit the manuscript page
- Figure 3 - it looks like the caption/explanation is missing?
- Lines 259-260: it should be also emphasised that HPV can cause much more harm, also among men. Now the reader might have an impression that it is women-only problem.
Author Response
first of all – congratulations for your great effort and interesting topic of the paper. I hope that my hints will be helpful and your final paper will be of great value to the scientific community. I really enjoyed reading it – let’s make it clear.
We thank the reviewer for their supportive comments.
- Despite the well-rooted habit of using abbreviations in Medicine sometimes too many such shortcuts make it more difficult to understand instead of facilitating the topic. It might be worth considering to only use well-known shortcuts, but not to create too many new ones.
We worked on all non-standard abbreviations, thank you for your attention to these details.
- In some paragraphs formatting should be improved, as well as double spaces etc.
We have made these changes. Please let us know if there are additional formatting issues we didn’t find.
- Chapter 2: the history of immunotherapy starts a bit earlier, it may be good to highlight it; there are several good reviews in this regard, e.g. 0.3389/fimmu.2019.02965
Thank you very much for the suggestion—this was an excellent paper!
Paragraph added in chapter 2: “Multiple anecdotal reports dating back to ancient Egypt and the early nineteenth century describe spontaneous tumor regression after an infection with concomitant high fever [1]. Two German physicians, Fehleisen and Busch, noticed significant tumor regression after erysipelas infection, making them the first to attempt to modulate patients' immune systems to cure cancer [2]. However, their attempts to replicate the phenomenon proved unsuccessful until Fehleisen identified the bacterial strain responsible for erysipelas and tumor shrinkage as Streptococcus pyogenes. The formal beginning of immunotherapy dates to 1891, when William Bradley Coley discovered 47 case reports of cancer patients undergoing spontaneous remission after concomitant acute bacterial infection [3].”
- Lines 72-72: the beginnings, ok, but why those early experiments were stopped? Why resumed decades later?
Additional sentences were added (in chapter 2) to address this: “Although "Coley's toxins" were commercially available from 1899 and proved successful, their lack of a known mechanism of action and the high risk of infecting cancer patients with pathogenic bacteria led oncologists to prefer surgery and radiotherapy [4, 5]. Furthermore, when researchers attempted to translate these findings to human trials, they faced significant challenges. The early trials were largely unsuccessful, and as a result, the idea of cancer immunotherapy was largely abandoned for several decade until the discoveries in the mid-twentieth century of key elements of the immune system, including interferon [6].”
- Line 86: I believe "increased RISK OF cancer development"?
These changes were made, thank you for the careful attention to detail.
- Figure 1: there are many more antibodies anti PD-1/PD-L1 currently, both FDA-approved and under clinical trials; it should be good (from a scientific point of view) to mention them, not solely pembro/atezo; this may suggest you're being paid by those pharma companies and we don't want to lose your scientific objectiveness.
Additional drugs/clinical trials were added, thank you for the suggestion.
- Figure 2 and 3 - size should be changed to fit the manuscript page.
These changes were made.
- Figure 3 - it looks like the caption/explanation is missing?
Caption was added to figure 3: “Figure 3. Cancer Vaccines Activate the Immune System to Target Tumor Cells. Cancer vaccines introduce tumor antigens (proteins found on the surface of cancer cells) into antigen-presenting cells (APCs), such as dendritic cells which then activate CD8+, CD4+, and B cells to mount an immune response against cancer cells. CD8+ T cells, also known as cytotoxic T cells, are responsible for killing cancer cells directly. CD4+ T cells, also known as helper T cells, help activate and coordinate the immune response by releasing cytokines that attract other immune cells to the site of the tumor. B cells, another type of immune cell, produce antibodies that can target cancer cells. Antibodies can bind to tumor antigens on the surface of cancer cells, marking them for destruction by other immune cells or activating complement proteins that lead to cell death.”
- Lines 259-260: it should be also emphasised that HPV can cause much more harm, also among men. Now the reader might have an impression that it is women-only problem.
Added in line ~305 mention of “penis” and 307-310: “It is important to note that HPV can affect both men and women, and as such, HPV vaccination is recommended for both boys and girls to help prevent the spread of the virus and reduce the risk of HPV-related cancers.”
Round 2
Reviewer 1 Report
The authors have answered all of my questions.
Reviewer 2 Report
Dear Authors,
congratulations for your great work, good luck in your further actions,
P.